# Microbiome and fragmentation pattern of blood cell-free DNA and fecal metagenome enhance colorectal cancer micro-dysbiosis and diagnosis analysis: a proof-of-concept study

Zhongkun Zhou,[1] Yunhao Ma,[1] Dekui Zhang,[2] Rui Ji,[3] Yiqing Wang,[3] Jianfang Zhao,[4] Chi Ma,[2] Hongmei Zhu,[1] Haofei Shen,[3] Xinrong Jiang,[1] Yuqing Niu,[1] Juan Lu,[1] Baizhuo Zhang,[1] Lixue Tu,[1] Hua Zhang,[1] Xin Ma,[1] Peng Chen[1]

ABSTRACT  Colorectal cancer (CRC) is the third most common cancer, and it can be prevented by performing early screening. As a hallmark of cancer, the human microbiome plays important roles in the occurrence and development of CRC. Recently, the blood microbiome has been proposed as an effective diagnostic tool for various diseases, yet its performance on CRC deserves further exploration. In this study, 133 human feces and 120 blood samples are collected, including healthy individuals, adenoma patients, and CRC patients. The blood cfDNA and fecal genome are subjected to shotgun metagenome sequencing. After removing human sequences, the microbial sequences in blood are analyzed. Based on the differential microbes and functions, random forest (RF) models are constructed for adenoma and CRC diagnosis. The results show that alterations of blood microbial signatures can be captured under low coverage (even at 3×). RF diagnostic models based on blood microbial markers achieve high area under the curve (AUC) values for adenoma patients (0.8849) and CRC patients (0.9824). When the fragmentation pattern is combined with microbial and KEGG markers, higher AUC values are obtained. Furthermore, compared to the blood microbiome, the fecal microbiome shows a different community composition, whereas their changes in KEGG pathways are similar. Pathogenic bacteria *Fusobacterium nucleatum* (*F. nucleatum*) in feces increased gradually from the healthy group to the adenoma and CRC groups. Additionally, *F. nucleatum* in feces and blood shows a positive correlation in CRC patients. Cumulatively, the integration of blood microbiome and fragmentation pattern is promising for CRC diagnosis.

IMPORTANCE  The cell-free DNA of the human microbiome can enter the blood and can be used for cancer diagnosis, whereas its diagnostic potential in colorectal cancer and association with gut microbiome has not been explored. The microbial sequences in blood account for less than 1% of the total sequences. The blood microbial composition, KEGG functions, and fragmentation pattern are different among healthy individuals, adenoma patients, and CRC patients. Machine learning models based on these differential characteristics achieve high diagnostic accuracy, especially when they are integrated with fragmentation patterns. The great difference between fecal and blood microbiomes indicates that microbial sequences in blood may originate from various organs. Therefore, this study provides new insights into the community composition and functions of the blood microbiome of CRC and proposes an effective non-invasive diagnostic tool.

KEYWORDS  microbiome, cell-free DNA, colorectal cancer, diagnosis, fragmentation pattern

**Peer Reviewer** Jun Hu, Tianjin Tumor Hospital, Tianjin, China

Address correspondence to Peng Chen, chenpeng@lzu.edu.cn.

The authors declare no conflict of interest.

See the funding table on p. 15.

Colorectal cancer (CRC) is the third common cancer in both sexes, accounting for about 10% of the estimated new cancer cases and deaths (1). Benefiting from early screening, morbidity and mortality of many cancers decreased during the last decades. However, the incidence of CRC increases in some developing countries, and many patients are diagnosed at the late stage, which causes a higher financial burden (2). Taking the low participation of colonoscopy and low sensitivity of fecal immunochemical test or multitarget stool DNA test for precancerous lesions into consideration, it is important to develop new diagnostic tools (3, 4).

The etiology of CRC is multifactorial, and polymorphic microbiomes have been proposed as a new hallmark of cancer (5). Among previous studies, diagnostic models based on microbial markers for CRC have achieved high sensitivity and specificity (6, 7). Recently, blood-based microbiome analyses for cancer diagnosis were proposed, providing a novel diagnostic tool. A randomized trial of patient adherence demonstrated that a blood-based screening test could promote CRC screening among adults who had declined prior CRC screening (8). Traditionally, blood is thought to be a sterile environment. Nevertheless, an increasing amount of evidence supports the existence of the blood microbiome. As early as 1969, Paparelli observed the bacterial forms in erythrocytes of healthy people (9). Subsequently, more researchers reported blood bacteria using light and electron microscopy, 16S rRNA sequencing, cell-free DNA (cfDNA) sequencing, and cell-free RNA (cfRNA) sequencing (10–13). After filtering out human reads, microbial sequences only account for approximately 1% of the total cfDNA. Over two-thirds of the sequences were bacterial, and they were verified by PCR amplification (10). Ding et al. performed a proof-of-concept study and found 127 differential species between CRC patients and healthy individuals (11). Moreover, patients with bacteremia from *Bacteroides fragilis*, *Streptococcus gallolyticus*, and *Fusobacterium nucleatum* (*F. nucleatum*) have an increased risk of CRC, indicating potential correlations between blood and gut microbiome (14). In terms of the origin of the blood microbiome, the current consensus is that they come from other organs instead of habituating in the blood all the time. Although the origin of the blood microbiome is still unresolved, it can be used for cancer diagnosis.

In 2019, Velculescu found that genome-wide cfDNA fragmentation in cancer patients was different from healthy people, and this kind of fragmented mode could be used to identify the tissue of origin of cancers (15). Afterward, fragmentation profiles and the microbiome of blood cfDNA were combined to improve the diagnostic performance. Wu et al. used human cfDNA fragments and microbial markers to identify sepsis and predict clinical outcomes in intensive care unit (16). Similarly, Wang et al. found that both human and microbial cfRNA had cancer-type specificity and improved the average recall by approximately 8% (12). Thereby, the combination of blood microbiome and fragmentation patterns can facilitate disease diagnosis.

In this study, blood cfDNA and fecal metagenomic sequencing were used to analyze microbiome dysbiosis. Diagnostic models were constructed based on microbes, microbial functions, and fragmentation patterns, providing a promising non-invasive screening strategy.

## RESULTS

### Clinical characteristics of the patients

Cohort 1 contained 120 blood samples, including 44 healthy individuals (BMGH), 31 adenoma patients (BMGP), and 45 CRC patients (BMGT). Cohort 2 contained 133 feces, including 55 healthy individuals (FMGH), 38 adenoma patients (FMGP), and 40 CRC patients (FMGT). Furthermore, 73 out of 120 fecal samples have matched blood samples, including 33 healthy individuals, 17 adenoma patients, and 23 CRC patients (Fig. 1A). No significant differences in age were observed among the three groups, yet gender differences existed. To eliminate the impact of gender, Pearson correlation analysis was performed, and then related microbes and functions were not included in the diagnostic models.

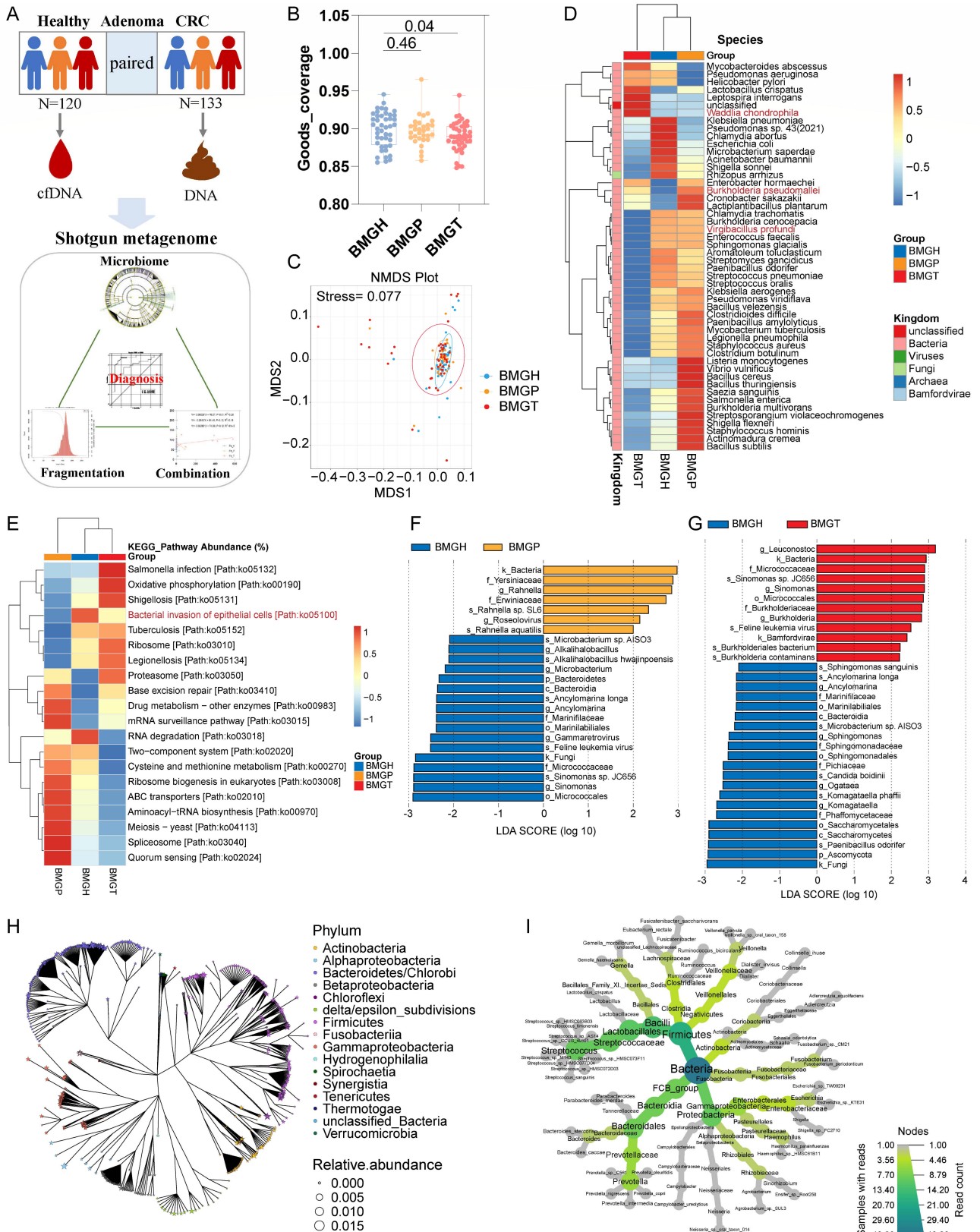

**FIG 1** Sample contorts and the microbiome analysis. Cohorts of the blood cfDNA and fecal genome shotgun metagenome sequencing (A); the alpha diversity among three groups. Data are mean ± SEM. *P* values were calculated by Wilcoxon test (B); the NMDS analysis at the phylum level (C); the relative abundance of the most abundant 50 species in three groups (microbes marked as red are correlated with gender according to Pearson correlation) (D); the relative abundance of the most abundant 20 KEGG pathways (E); LEfSe analysis between BMGH and BMGP groups (F); LEfSe analysis between BMGH and BMGT groups

Fig 1 (Continued)

(G); and phylogenetic tree of blood microbiome. The blood sample types include colon adenocarcinoma, esophageal carcinoma, head and neck squamous cell carcinoma, rectum adenocarcinoma, and stomach adenocarcinoma. The number of blood samples is 401 (H); the phylogenetic tree of the most abundant 40 species in the blood of CRC patients. The number of CRC blood samples is 141 (I).

## Differences in blood microbiome between healthy people and patients

The blood microbiome sequencing results showed that the microbial reads accounted for less than 1% of the total reads after filtering out human sequences, which is consistent with a previous study (10). The alpha diversity gradually decreased from the healthy group to the CRC group ($P = 0.04$; Fig. 1B). This kind of change was also reported in the fecal microbiome (17). Non-metric multidimensional scaling (NMDS) analysis of the three groups exhibited differences at the kingdom level (Fig. 1C). At the phylum level, Proteobacteria (10%), Firmicutes (7%), and Actinobacteria (2%) were the most abundant phyla. At the species level, *Escherichia coli*, *Staphylococcus aureus,* and *Streptococcus pneumoniae* were the most abundant species (Fig. S1A through C). In the healthy group, *Klebsiella pneumoniae*, *Pseudomonas* sp. 43, *Chlamydia abortus,* and *E. coli* were more abundant, whereas *Listeria monocytogenes*, *Vibrio vulnificus*, *Bacillus cereus,* and *Bacillus thuringiensis* were more abundant in adenoma patients, and *Mycobacteroides abscessus*, *Leptospira interrogans*, and *Helicobacter pylori* were the dominant species in CRC patients (Fig. 1D). Compared to the BMGH group, RNA degradation was less abundant in BMGP and BMGT, whereas *Salmonella* infection and oxidative phosphorylation were increased in BMGT, and quorum sensing and aminoacyl-tRNA biosynthesis were enriched in BMGP (Fig. 1E). Next, the linear discriminant analysis effect size (LEfSe) analysis was performed. Compared to healthy individuals, Yersiniaceae, *Rahnella,* and Erwiniaceae were enriched in adenoma patients, whereas *Micrococcales*, *Sinomonas,* and fungi were decreased (Fig. 1F). In CRC patients, *Leuconostoc*, *Sinomonas* sp. JC656, and Burkholderiaceae were enriched, but fungi, *Ascomycota*, *Paenibacillus odorifer,* and *Saccharomycetes* were depleted (Fig. 1G). Furthermore, the dominant species in CRC was verified by blood whole genome sequencing from 141 CRC samples in the TCMA database (Fig. 1H and I) (18). Although the blood microbiome is different from the fecal microbiome reported by previous studies (7), the disorder of microbial composition provides a basis for CRC diagnosis.

It is acknowledged that the human microbiome is affected by many factors and is different in different individuals, whereas their functions are similar (6, 19, 20). Compared to the BMGH group, the top 10 differential KEGG pathways between BMGH and BMGD (BMGP and BMGT) groups are enriched in the BMGD group. In particular, the bacterial secretion system was significantly upregulated in BMGD (Fig. S1D), indicating that the blood microbial cfDNA might originate from bacterial extracellular vesicles.

## Combination of blood microbiome and fragment size improves diagnostic performance

It was reported that the fragmentation pattern of blood cfDNA could be used for diagnosis and identification of tissue origin (21, 22), and then the fragmentation pattern was analyzed. The length of cfDNA was mostly distributed between 150 and 200 bp. The density curves revealed that the BMGH and BMGP curves are closer to the right side (Fig. 2A). Next, the insert size peak of each sample was selected, and it showed that the cfDNA in BMGT was shorter and was significantly different from BMGH (Fig. 2B), which could be helpful for diagnosis.

After obtaining the microbial composition and function differences, classification models were constructed. Taking the relatively high accuracy and evaluability of the importance of each marker into consideration, random forest (RF) was the most common algorithm (7). The potential markers were firstly selected by the AUCRF package, and then a panel (BMGH vs BMGP) including 15 microbes obtained an area under the curve

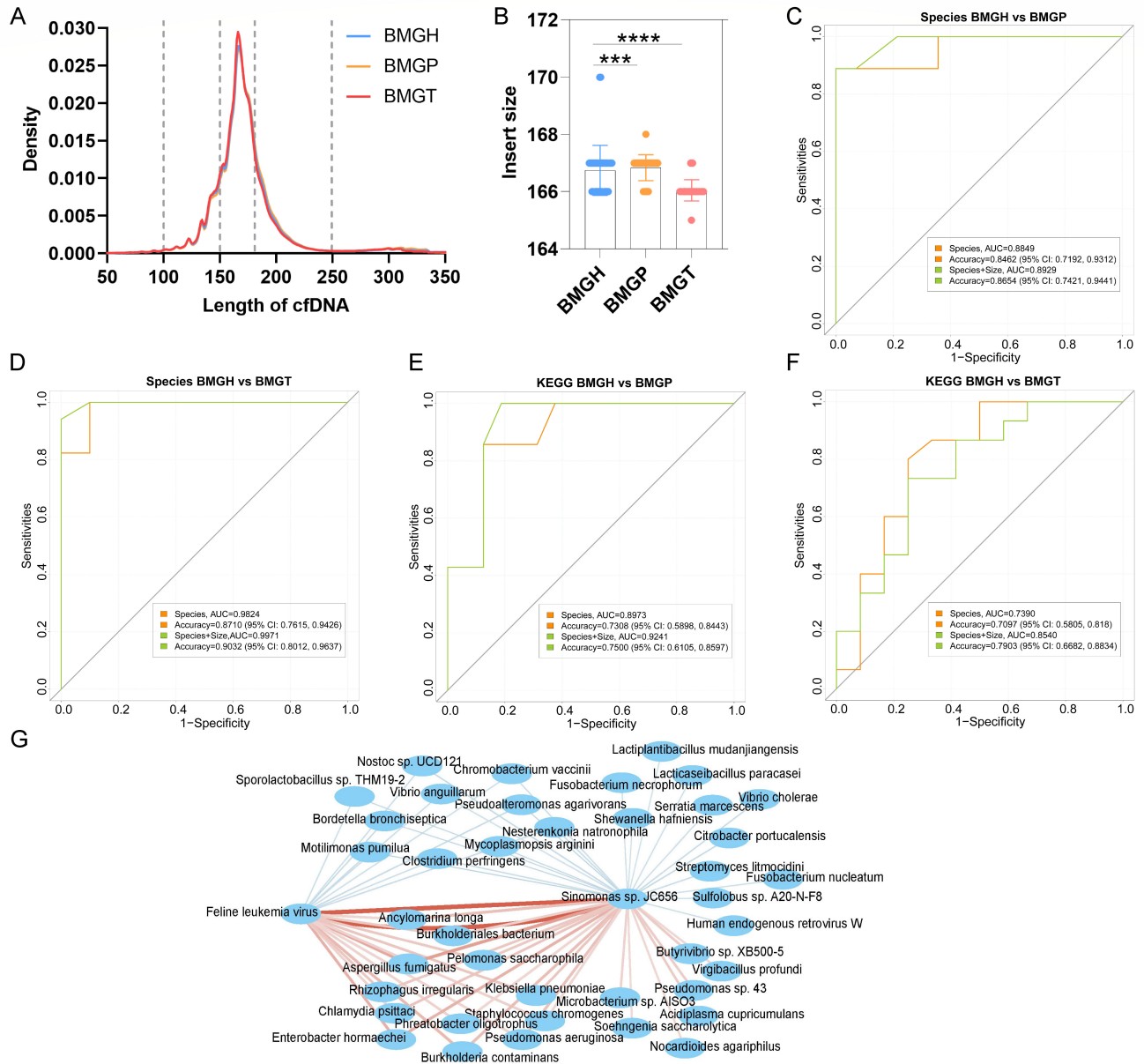

**FIG 2** The diagnostic performance based on microbiome markers and fragment patterns. The density curves based on cfDNA length (A); the insert size peak of each sample. Data are mean ± SEM (*t*-test, ***$P < 0.001$; ****$P < 0.0001$) (B); AUC curves between BMGH and BMGP based on microbial markers and fragmentation length (C); AUC curves between BMGH and BMGT based on microbial markers and fragmentation length (D); AUC curves between BMGH and BMGP based on KEGG pathways and fragmentation length (E); AUC curves between BMGH and BMGT based on KEGG pathways and fragmentation length (F); Species that are correlated with *Feline leukemia virus* and *Sinomonas* sp. JC656 ($P < 0.05$, red lines represent positive correlations, blue lines represent negative correlations, and the line thickness is correlated with the absolute value of coefficient) (I).

(AUC) of 0.8849 (accuracy, 0.8462; 95% CI: 0.7192–0.9312; sensitivity, 0.8182; and specificity, 0.8667) (Fig. 2C). When the fragment size information was added, the AUC value increased to 0.8929 (accuracy, 0.8654; 95% CI: 0.7421–0.9441; sensitivity, 0.8571; and specificity, 0.8710). Similarly, a panel (BMGH vs BMGT) including nine microbes obtained an AUC of 0.9824 (accuracy, 0.8710; 95% CI: 0.7615–0.9426; sensitivity, 0.8333; and specificity, 0.9062) (Fig. 2D). When the fragment size information was added, the AUC value increased to 0.9971 (accuracy, 0.9032; 95% CI: 0.8012–0.9637; sensitivity, 0.8929; and specificity, 0.9118). Afterward, KEGG pathways were used as diagnostic markers. A panel of five altered functions (oxidative phosphorylation, pertussis, carbon

fixation, base excision repair, and glutathione metabolism pathways) obtained an AUC of 0.8973 (accuracy, 0.7308; 95% CI: 0.5898–0.8443; sensitivity, 0.7083; and specificity, 0.7500) for BMGP diagnosis. After the addition of fragment size, the AUC value increased to 0.9241 (accuracy, 0.7500; 95% CI: 0.6105–0.8597; sensitivity, 0.7391; and specificity, 0.7586) (Fig. 2E). Similarly, six KEGG pathways (oxidative phosphorylation, nucleocytoplasmic transport, alanine aspartate and glutamate metabolism, glycolysis and gluconeogenesis, shigellosis, and atrazine degradation pathways) obtained an AUC of 0.7390 (accuracy, 0.7097; 95% CI: 0.5805–0.8180; sensitivity, 0.7692; and specificity, 0.6667) for BMGT diagnosis. When the fragment size was added, the AUC value increased to 0.8540 (accuracy, 0.7903; 95% CI: 0.6682, 0.8834; sensitivity, 0.8276; and specificity, 0.7576) (Fig. 2F). Therefore, the combination of microbes and fragment size is a promising strategy for CRC diagnosis. Finally, the two most important species shared by RF models were selected to analyze the interactions with other microbes. *Sinomonas* sp. JC656 and *Feline leukemia virus* were enriched in BMGP and BMGT. They showed significantly positive correlations with common pathogenic microorganisms, such as *Chlamydia psittaci*, *K. pneumoniae*, *Enterobacter hormaechei,* and *Staphylococcus chromogenes*. Contrarily, *Lacticaseibacillus paracasei* was negatively correlated with *Sinomonas* sp. JC656, which provided insights for further microbiome regulation (Fig. 2G).

## Diagnostic models based on fecal microbiome

The microbiome of 133 feces was analyzed. The results showed that Bacteroidota, Bacillota, Pseudomonadota, Uroviricota, and Actinomycetota were the most abundant five phyla, accounting for more than 90% of the microbial community (Fig. 3A). *Bacteroides*, *Phocaeicola*, *Escherichia*, *Faecalibacterium,* and *Segatella* were the most abundant genera (Fig. 3B). LEfSe results demonstrated that Bacilli, *Porphyromonas*, Porphyromonadaceae, Actinomycetota, and viruses were enriched in CRC patients, and the abundance of other harmful microbes, such as *Methanobrevibacter smithii*, *Porphyromonas asaccharolytica*, *F. nucleatum,* and *Gemella morbillorum*, was also increased. In adenoma patients, *Phocaeicola*, *Phocaeicola vulgatus*, Bacteria, *Fusobacterium hominis*, *Parabacteroides distasonis,* and *Bacteroides* were more abundant (Fig. 3C). Compared to FMGH, ribosome, quorum sensing, peptidoglycan biosynthesis, pyrimidine metabolism, and lysine degradation were more abundant in FMGT, whereas lipopolysaccharide biosynthesis, pentose and glucuronate interconversions, arginine and proline metabolism, fructose and mannose metabolism, and biotin metabolism were enriched in FMGP (Fig. 3D). In particular, *Phocaeicola* and *Bacteroides* in FMGP have abundant glycoside hydrolases, which may promote fructose and mannose metabolism (Fig. 3D and E). This function in FMGH was mainly kept by *Faecalibacterium*, *Roseburia*, *Ruminococcus,* and *Clostridium*, which reflected the differences in microbial composition and their functions between healthy people and patients (Fig. 3E). The RF model showed that eight microbial markers for FMGP diagnosis and eleven microbes for FMGT diagnosis obtained AUC values of 0.8877 (accuracy, 0.7385; 95% CI: 0.6146–0.8397; sensitivity, 0.7083; and specificity, 0.7561) and 0.8806 (accuracy, 0.8030; 95% CI: 0.6868–0.8907; sensitivity, 0.8214; and specificity, 0.7419), respectively (Fig. 3F). Moreover, fifteen differential KEGG pathways between FMGP and FMGH and fifteen differential pathways between FMGT and FMGH obtained AUC values of 0.7708 (accuracy, 0.7231; 95% CI: 0.5981–0.8269; sensitivity, 0.7000; and specificity, 0.7333) and 0.9608 (accuracy, 0.7727; 95% CI: 0.6530–0.8669; sensitivity, 0.7222; and specificity, 0.7917), respectively (Fig. 3G).

## Differences between matched blood and fecal microbiome

Next, fecal samples that were matched with blood were selected. The top ten differential species were *Eggerthia catenaformis*, *Eikenella* sp. HMSC061C02, *Fusobacterium* sp. Oral taxon 203, *Lachnoanaerobaculum orale*, *Lactobacillus gasseri*, *Peptoniphilus catoniae*, *Peptoniphilus* sp., Marseille Q7072, *Porphyromonas* sp. CAG:1061, and uncultured *Parvimonas* sp. (Fig. 4A). The most abundant metabolic pathways were proteasome, mineral absorption, lysine degradation, tuberculosis, and sphingolipid metabolism in the

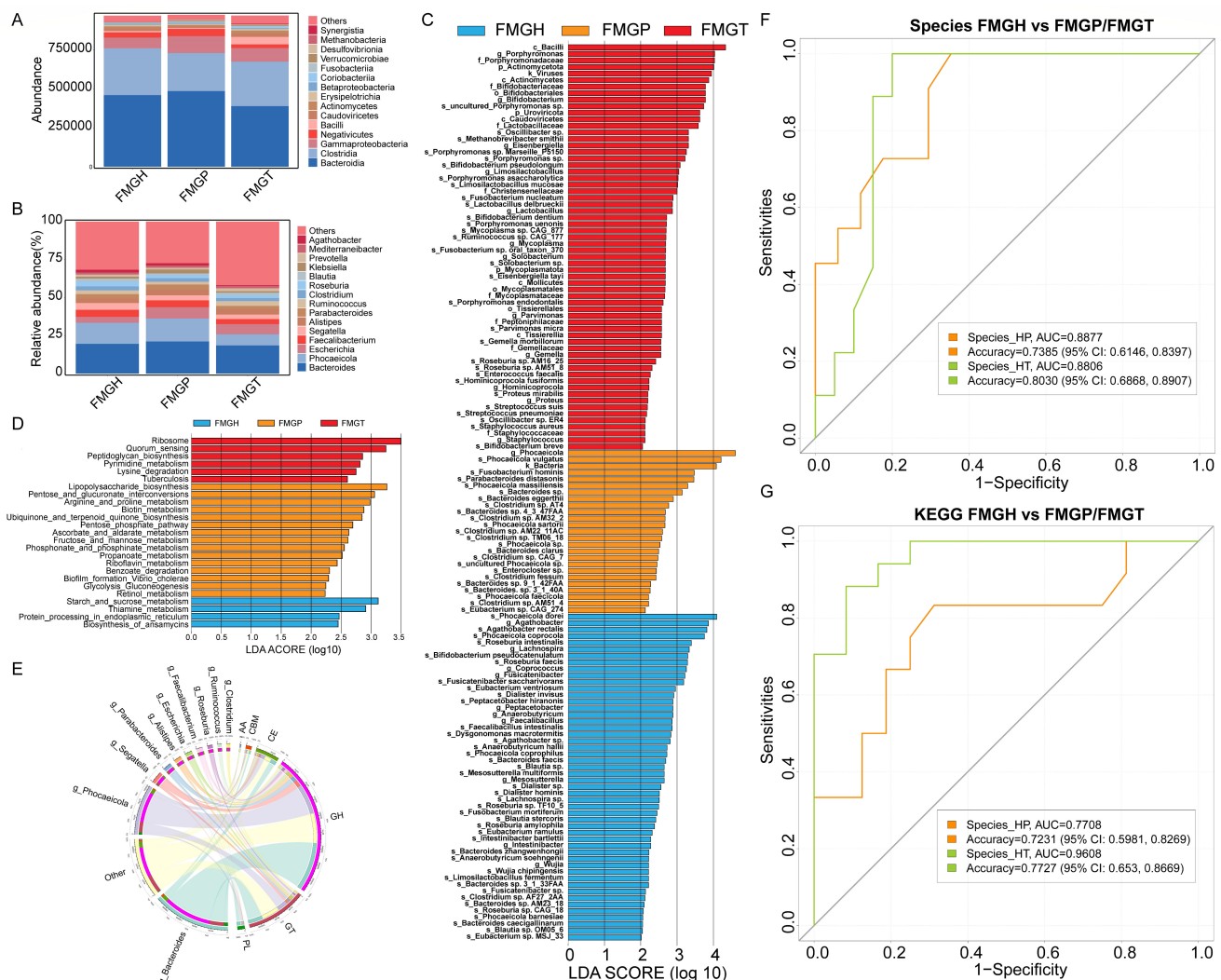

**FIG 3** The analysis of fecal microbiome. The community composition at the phylum level (A) and at the genus level (B); LEfSe analysis of species (C) and KEGG pathways (D) among FMGH, FMGP, and FMGT groups; Functions of the dominant genera (E); AUC curves based on differential species (F); and AUC curves based on differential KEGG pathways (G).

FMGT group (Fig. 4B). Thus, the differential microbes in feces were different from those in blood. Nevertheless, the differential functions showed similarities, and it suggested that different microorganisms were involved in similar functions in CRC patients. For example, the upregulated oxidative phosphorylation in blood microbiome and glycolysis/sphingolipid metabolism in the fecal microbiome are important for energy production in CRC.

Based on these matched fecal samples, diagnostic models were constructed. For FMGP diagnosis, a panel of nine bacteria obtained an AUC of 0.7045 (accuracy, 0.8286; 95% CI: 0.6635–0.9344; sensitivity, 0.8182; and specificity, 0.8333) (Fig. 4C). For FMGT diagnosis, seven bacteria obtained an AUC of 1.0000 (accuracy, 0.8718; 95% CI: 0.7257–0.9570; sensitivity, 0.9375; and specificity, 0.8261) (Fig. 4C). Next, KEGG annotations were used for model construction. Nine differential KEGG pathways between FMGP and FMGH and five differential pathways between FMGT and FMGH obtained AUC values of 0.7045 (accuracy, 0.8571; 95% CI: 0.697–0.9519; sensitivity, 0.7857; and specificity, 0.9048) and 0.7286 (accuracy, 0.8205; 95% CI: 0.6647–0.9246; sensitivity, 0.8462; and specificity, 0.8077), respectively (Fig. 4D). From the perspective of function alterations, FMGP is different from FMGT, which needs specific biomarkers.

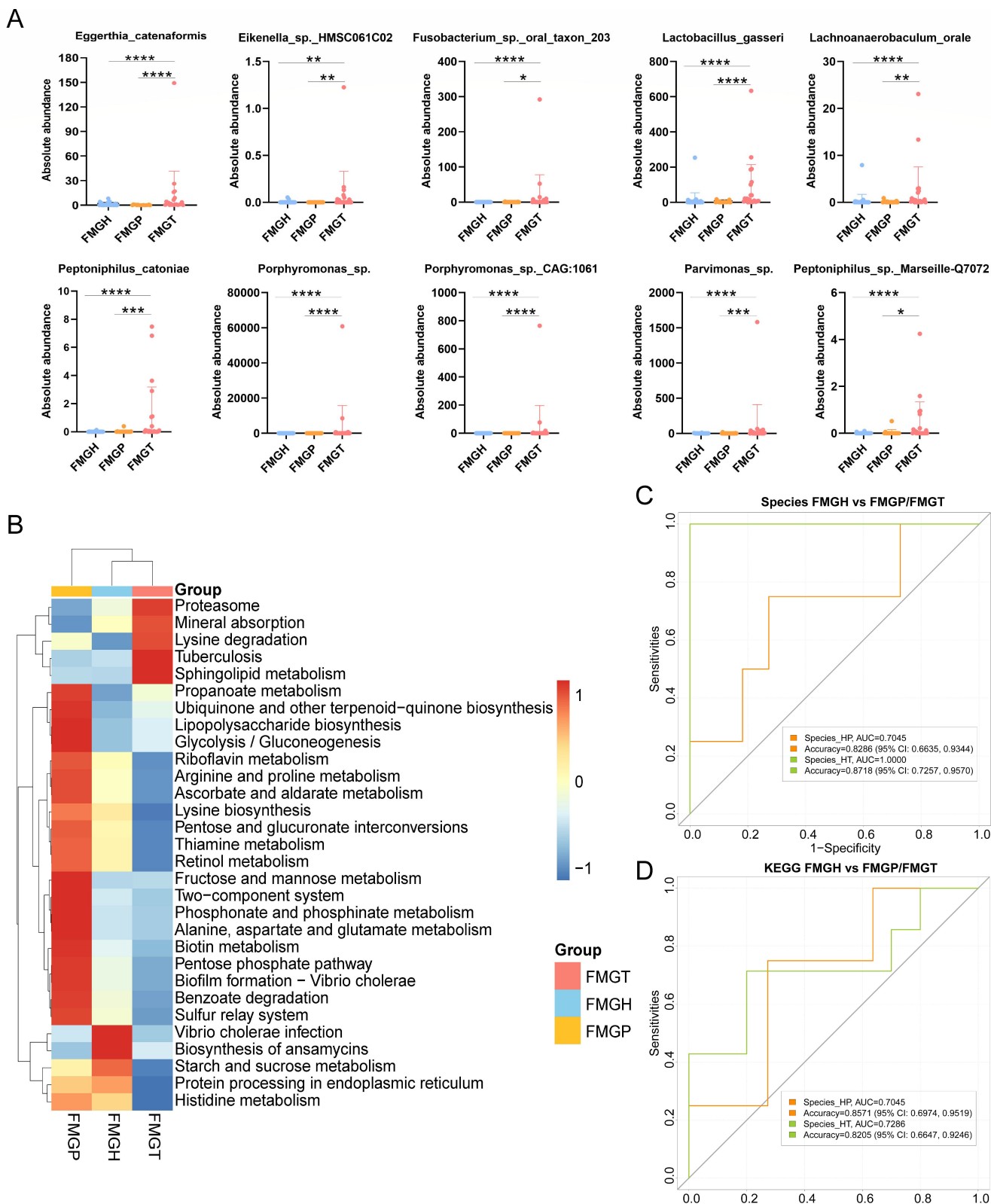

**FIG 4** The microbiome analysis of matched fecal samples. The top 10 differential species among the three groups. Data are mean ± SEM (*t*-test, *P < 0.05; **P < 0.01; ***P < 0.001; ****P < 0.0001) (A); the most abundant 30 KEGG pathways in three groups (B); AUC curves based on differential species (C); and AUC curves based on differential KEGG pathways (D).

## Interactions between matched fecal and blood microbiome

Considering the difference of microbial biomarkers and partial similarity of KEGG markers, matched blood and fecal samples were selected for further analysis. There were 177 species shared by blood and fecal microbiome. The Pearson correlations were calculated (Fig. S2A and B). Finally, species with *P* values lower than 0.05 were selected, and only eleven species showed significant correlations (eight positive correlations and three negative correlations, Fig. 5A). Among these microbes, *Clostridioides difficile* (*C. difficile*) is a common pathogenic bacterium, which drives colonic tumorigenesis through TcdB toxin (23). Although no significant differences were observed between healthy individuals and patients (Fig. 5B and C), *C. difficile* might translate from gut to blood, which needs further verification. In terms of the importance of *F. nucleatum* in the occurrence, development, and diagnosis of CRC, it was selected for verification. In the fecal samples, its abundance increased gradually from the healthy group to the adenoma and CRC groups (Fig. 5D), which is consistent with the previous study (24). However, this kind of difference lost statistical importance in blood microbiome, although there was a gradually increasing trend (Fig. 5E). Interestingly, when the blood samples were divided into three groups, *F. nucleatum* showed a positive correlation between blood and fecal samples in the CRC group, with *P* values being decreased from 0.26 to 8*e−5 (Fig. 5F). Thus, it suggested that more cfDNA of *F. nucleatum* entered the blood with the development of CRC. However, the detailed pathway, such as intestinal epithelial injury, blood transmission, or bacterial exocrine vesicles, requires further exploration.

## Disorder of the blood microbiome can be captured under low sequencing depth

It was reported that altered microbiome and fragmentation patterns under high sequencing depth (30×) could also be observed under 1× sequencing depth (11, 15). Hence, 18 blood samples were used to verify the influence of sequencing depths (3×

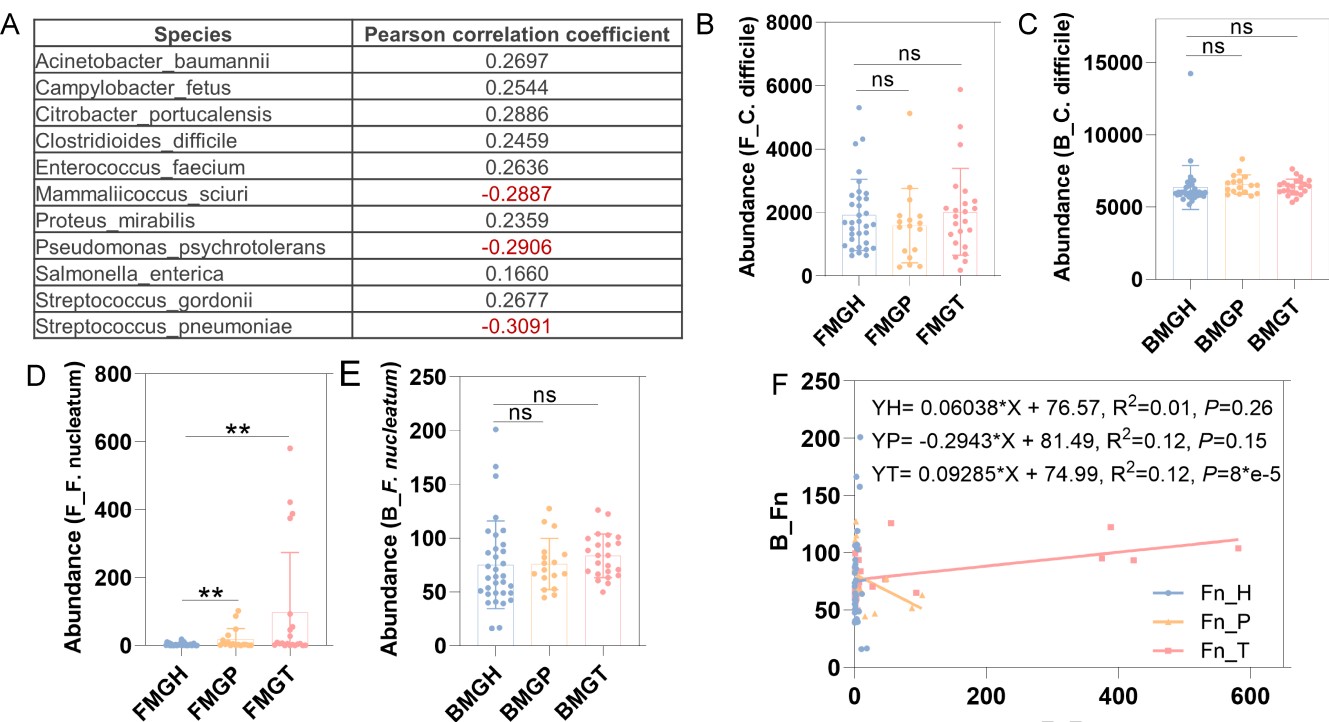

**FIG 5** Interactions between blood and fecal microbiome. The shared 11 species in feces and blood. *P* values of the Pearson correlation are lower than 0.05 (A); the absolute abundance of *C. difficile* in feces and blood. Data are mean ± SEM (*t*-test, ns, *P* > 0.05) (B, C); the absolute abundance of *F. nucleatum* in feces and blood. Data are mean ± SEM (*t*-test, ns, *P* > 0.05; **, *P* < 0.01) (D, E); Correlations between feces and blood of *F. nucleatum* in healthy, adenoma, and CRC groups (F).

and 6×). In terms of alpha diversity, no significance was observed (Fig. 6A). Similarly, the beta diversity showed a more apparent separation among H, P, and T groups than groups of different sequencing depths (3× and 6×) (Fig. 6B). The Pearson correlation was higher than 0.95 between the same sample under different sequencing depths, whereas correlation coefficients were much lower among H, P, and T groups (Fig. 6C). Thus, the influence of disease classification on microbiome was larger than that of sequencing depth (Fig. 6D). In addition, this kind of influence was verified by community composition and KEGG annotations (Fig. 6E and F). Furthermore, only limited microbes showed differences between 3× and 6×, and they did not belong to the diagnostic markers (Fig. 6G through I). Therefore, lower sequencing depth for blood microbiome-based diagnosis is promising.

## DISCUSSION

Here, blood and fecal samples were collected for shotgun metagenome sequencing, and then the differential microbes, functions, and fragmentation patterns were used for CRC diagnosis. Furthermore, matched fecal samples were selected to analyze the interactions between fecal microbiome and blood microbiome. Importantly, this study firstly combined fragmentation patterns and blood microbiome for CRC diagnosis, and this integration achieved high accuracy.

Different parts of the blood contain different microbes, and most of the blood bacterial DNA is in buffy coat (93.74%). Red blood cells contain more bacterial DNA (6.23%) than the plasma (0.03%), with Proteobacteria, Actinobacteria, Firmicutes, and Bacteroidetes ranking as the most abundant phyla (25). Different diseases have different biomarkers, but the reports about blood microbiome for CRC and adenoma diagnosis are limited. cfRNA sequencing results demonstrated that Proteobacteria, Firmicutes, Actinobacteria, Bacteroidetes, and Cyanobacteria were the dominant phyla in CRC blood (12). However, there were no detailed analyses for functional changes. Our results showed that energy metabolism, translation, and infectious disease pathways were increased in CRC patients, which might reflect the destruction of intestinal epithelium and transfer of microbes. In adenoma patients, functional changes included enhanced amino acid metabolism, lipid metabolism, and glycan biosynthesis and metabolism. Therefore, previous 16S rRNA sequencing-based analysis of the blood microbiome may not be effective enough for diagnosis.

Fecal microbiome has been studied by numerous researchers. Except for the bacteria, mycobiome and virome were also analyzed, revealing the complex multi-kingdom interactions (26–28). In our study, there were about 300 species detected in blood, while almost 20,000 species were identified in feces. Furthermore, 177 species were found in both of them, but only 11 species showed significant correlations, indicating that other organs might make contributions to blood microbial cfDNA. In terms of KEGG changes, alterations in fecal microbiome were related to lysine degradation and sphingolipid metabolism, and the functional changes in blood microbiome were associated with oxidative phosphorylation. Moreover, some harmful microbes, such as *F. nucleatum*, are enriched in both feces and blood of CRC patients. According to previous studies, the nucleic acid of these microbes might enter blood circulation through disruption of the intestinal mucosal barrier, extracellular vesicle secretion, bleeding gums, and so on. As reviewed by D'Alessandra, extracellular vesicles generated by prokaryotes could shuttle different intracellular components, such as proteins and nucleic acids, and they could be found in human biofluids (29). Previous studies also demonstrated that viral signatures detected in microbial cfDNA in plasma had clinical correlations, including clinical efficacy of empirical antibiotic treatment, progression to acute-on-chronic liver failure, and short-term mortality (30). Therefore, clinical management may benefit from blood microbiome analysis.

*F. nucleatum* is a common oral microorganism, and it was found to be enriched in CRC in 2012 (31). Moreover, *F. nucleatum* was found to be enriched in the feces, saliva, and tumor tissues (32). Our results showed that it was also more abundant in the blood

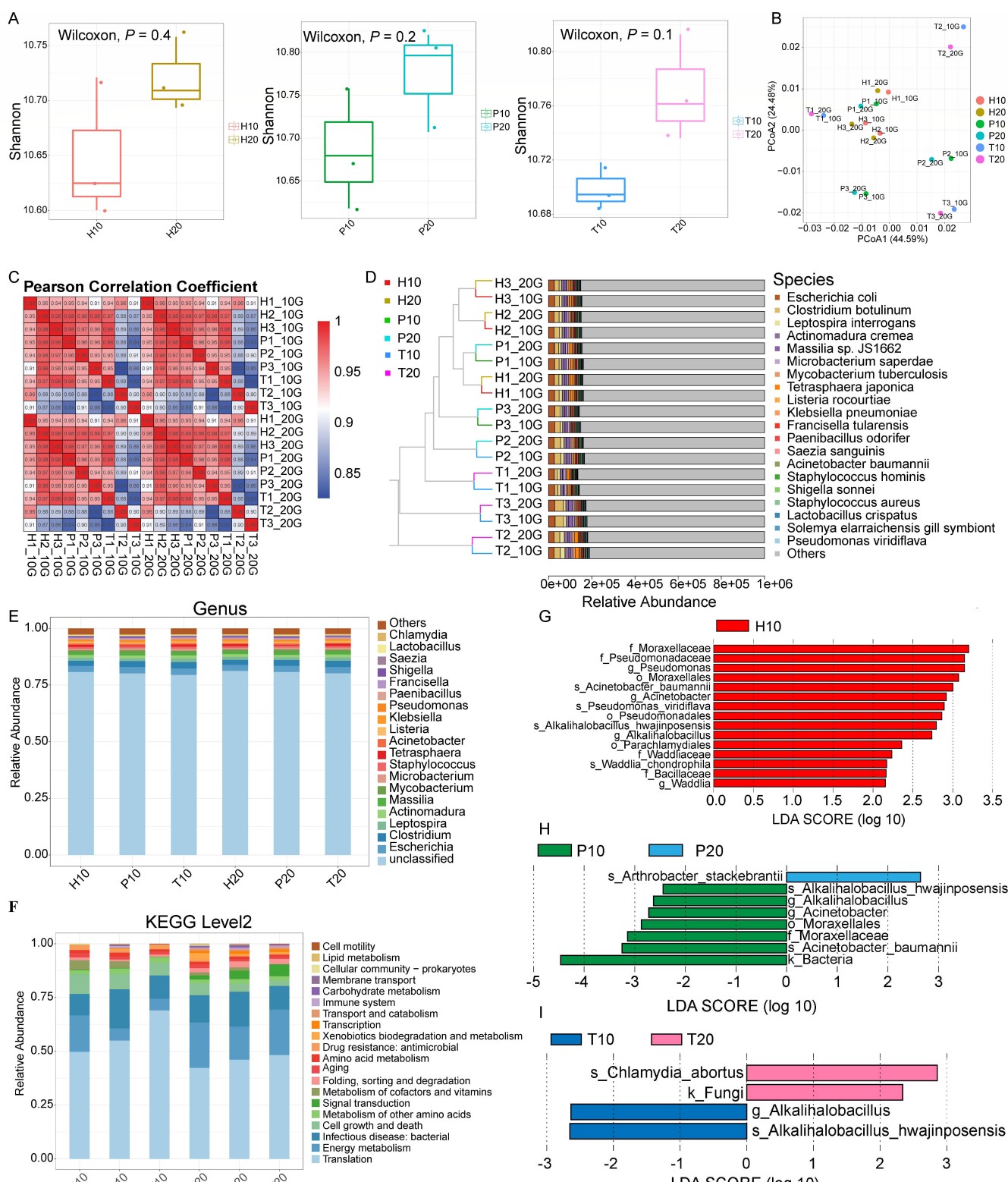

**FIG 6** Evaluations of metagenome results under 3× and 6× depths. Shannon index among H, P, and T groups. Data are mean ± SEM. *P* values were calculated by Wilcoxon test (A); PCoA at the genus level (B); Pearson correlations based on gene numbers (C); the unweighted pair-group method with arithmetic means (UPGMA) analysis at the species level (D); community composition at the species level (E) and composition of KEGG pathways (F); and LEfSe analysis of the three groups under different sequencing depth (G–I).

microbiome of CRC patients. In our diagnostic models based on blood microbiome, the oxidative phosphorylation pathway was upregulated in CRC patients. In the diagnostic models based on fecal microbiome, the upregulated pathways in CRC patients include lysine biosynthesis, arginine and ornithine metabolism, fatty acid biosynthesis, sphingolipid metabolism, and so on. It is well known that adenosine triphosphate (ATP) production by oxidative phosphorylation is 18 times more than glycolysis, and both of them are important for the proliferation and tumorigenesis of CRC cells (33). Zhou reported that *F. nucleatum* infection mediated elevation of angiopoietin-like 4 (ANGPTL4) expression, which promoted glucose uptake and glycolysis activity in CRC cells. This process is necessary for the colonization of *F. nucleatum* in CRC (34). Fang found that *F. nucleatum* supported carcinogenesis via increasing CRC cell glucose metabolism. It targeted lncRNA ENO1-IT1 to promote glycolysis and oncogenesis in CRC (35). Duan revealed that *F. nucleatum* activated glycolysis, inhibition of which by 2-deoxyglucose in turn suppressed *F. nucleatum*-induced lipogenesis (36). An important feature of CRC cells is their ability to rewrite the fatty acid metabolism for ATP production and cell proliferation. Fatty acids are the building blocks for phospholipids, sphingolipids, and triglycerides (37, 38). *F. nucleatum* infection could induce lipid accumulation and FASN upregulation (39). Therefore, microbial dysbiosis can induce metabolic disorders and then promote CRC development.

Nowadays, it is found that molecular features of cfDNA are helpful for diagnosis and elucidation of tissue origin (40). In other diseases, integration of microbial and host information, such as fragmentation pattern, gene expression, and fragment-end composition, improved the prediction of sepsis, cancer types, cancer survival, and recurrence (12, 16, 41). Our models also confirmed some of these results. When the fragmentation information was included, the AUC values increased. Circulating DNA consists of a mixture of DNA from multiple tissues within the body. Tumor-derived DNA in cancer patients (1989) opened up the door of cfDNA-based noninvasive cancer liquid biopsies (22). cfDNA contains various molecular features characteristic, such as methylation, fragment size, end motif frequency, jagged end length, oriented end density, and nucleosomal footprinting (40). Through systematic evaluation of cfDNA fragmentation patterns, it was found that these regions were significantly enriched for cancer-related functional ontologies, such as the estrogen signaling pathway, Rab GTPase binding, lipid metabolic processes, and cytidine deaminase activity (42). Based on cell-free multi-omics analysis (cfNDA and cfRNA), Lu reported the cancer gene expression, immune signatures, related pathways, and origins of these cfRNA-derived signatures. For example, oxidative phosphorylation and PI3K-Akt signaling pathways were upregulated in cancer patients (43). In terms of the relationship between microbes and fragmentation patterns, previous studies found that *F. nucleatum* infection induced apoptotic cell death in peripheral blood mononuclear and polymorphonuclear cells. For example, it induced significant DNA fragmentation of Jurkat T cells (44). Moreover, *F. nucleatum* can promote DNA damage and is correlated with microsatellite instability (45). Therefore, the infection of *F. nucleatum* may contribute to the characteristic (shorter cfDNA length) of cfDNA in CRC patients.

Previous study used 25–30× whole-genome sequencing to analyze blood microbiome, and the bacterial markers were validated under 1× sequencing depth (11). The altered fragmentation patterns from cancer patients could be identified even at 0.5× coverage (15). 6× sequencing depth was chosen in this study, and higher coverage represents higher costs and computing resources. Recently, another study analyzed blood microbiome under 5× sequencing depth and constructed a cmDNA model for early diagnosis and recurrence of lung cancer (46). Under 6× conditions, the cost can be controlled within $200, and lower sequencing depth for diagnosis deserves exploration in the future.

There are several limitations to this study. First, although the blood microbiome was analyzed, the dynamic changes of the microbiome and fragmentation patterns during the development of CRC were still unclear, which needs to be resolved by longitudinal

studies. Second, the matched fecal sample cohort is not large enough, and meta-analysis will be a practical method to improve the robustness of our results. Third, in light of the diagnostic costs, the sequencing depth is relatively low, and other cfDNA characteristics were not explored. In the future, adding methylation or mutation signatures into the model to construct multi-modal models can facilitate the diagnosis of multifactorial disease. Finally, the causal relationship between microbial markers and CRC is still unclear. Rigorous mechanism studies based on Koch's postulates are supposed to be performed before clinical application (4, 47).

## Conclusion

In summary, metagenomic sequencing of blood and fecal microbiome was performed to analyze the alterations of diversity, community composition, and functions. After the integration of microbiome (microbes and related functions) and fragmentation patterns, better diagnostic performance was obtained. Eventually, results of matched fecal samples revealed that blood microbiome and gut microbiome are quite different, and features of blood microbiome could also be captured under low sequencing depth. Thus, multi-modal analysis based on blood cfDNA is a promising CRC diagnostic tool.

## MATERIALS AND METHODS

### Sample collection and storage

The fecal and blood samples were collected from patients in the Second Hospital of Lanzhou University, the First Hospital of Lanzhou University, and the Third People's Hospital of Gansu Province. The exclusion criteria are as follows: (i) younger than 18 years old; (ii) have antibiotics for treatment in the last 3 months; (iii) on a vegetarian diet; (iv) experiencing surgeries in the last 3 months; (v) inflammatory bowel disease and diarrhea; (vi) have any other cancer history; (vii) have received chemotherapy and radiation treatments; and (viii) have any infectious diseases. All the patients were confirmed by colonoscopy and pathological examination. Feces were collected before colonoscopy. Blood samples were collected using blood collection tubes containing EDTA. Blood samples were centrifuged at 3,000 rpm for 10 min, and then the supernatant was transferred to sterile tubes and stored at −80°C. The study protocol was approved by the Regional Ethical Review Board, and informed consents were obtained from each patient.

### DNA extraction, quality control, and library construction

cfDNA was extracted with MagPure Circulating DNA Mini Precast Kit (Guangzhou Magen Biotechnology Co., Ltd.) following the manufacturer's instruments. The quality of cfDNA was confirmed by Qubit 3.0 and Agilent 4200. Samples with a concentration lower than 10 ng/µL were discarded. Nuclease-free water was set as a negative control. The libraries were constructed using Rapid Plus DNA Lib Prep Kit for Illumina V2 (RK20255).

The fecal DNA was extracted with QIAamp Fast DNA Stool Mini Kit (Qiagen, Hilden, Germany) and DNA concentration was assessed by Nanodrop2000. The DNA was fragmented by S220 Focused Ultrasonicators (Covaris, USA) and cleaned up by Agencourt AMPure XP beads (Beckman Coulter Co., USA). The libraries were constructed using the TruSeq Nano DNA LT Sample Preparation Kit (Illumina, USA).

### Shotgun metagenome sequencing of cfDNA and fecal genomic DNA

The metagenome sequencing was performed on the Illumina Novaseq 6000 platform and 150 bp paired-end reads were generated. The raw data were trimmed and filtered using fastp software (48). The post-filtered pair-end reads were aligned against the

human genome using bbmap, and the metagenome assembly was performed using MEGAHIT (49).

In terms of cfDNA metagenome sequencing, reads were mapped to human genome version 19 using Bowtie2 to remove human sequences. Clean data were assembled using MEGAHIT, and then Bowtie2 was used to map the clean data of each sample to the assembled contigs. Next, unused paired-end reads were obtained to form a mixed sample for mixed assembly. QUAST was used to evaluate the assembly results. Prodigal was used to predict coding sequences (CDS) from contigs of single sample and mixed assembly, and sequences with CDS length less than 100 nt were filtered out. Subsequently, based on the CDS prediction results, CD-HIT was used for clustering, and the longest gene was selected as the representative sequence (Unigenes).

In terms of fecal genomic DNA metagenome sequencing, gaps inside scaffolds were used as breakpoints to interrupt the scaffold into new contigs (Scaftig), and Scaftig longer than 500 bp were retained. Open reading frames (ORFs) prediction of assembled scaffolds was performed using Prodigal and translated into amino acid sequences (50). The non-redundant gene sets were built for all predicted genes by MMSeqs2. The longest gene was selected as the representative sequence.

## Bioinformatics analysis of microbial composition

Clean reads of each sample were aligned against the non-redundant gene set (95% identity) using Salmon, and the abundance of the gene in the corresponding sample was calculated (transcripts per kilobase million [TPM]). Unigenes were aligned with the NR_meta library using DIAMOND. The result with an $E$ value $\leq$e*10 was selected for taxonomy identification. Annotation for each taxonomy of the sequence was performed using MEGAN, and the taxonomy includes kingdom, phylum, class, order, family, genus, and species.

## Functional annotation analysis

The gene set of representative sequences was mapped to Uniprot, KEGG, COG, CAZy, PHI, and VFDB databases with an $E$ value of 1e−5 using DIAMOND. The KEGG pathways were analyzed using KOBAS. The gene set was aligned with the CAZy database using hmmscan to obtain the carbohydrate enzyme. Unigenes were aligned with the protein sequences in the CARD database using RGI, and the annotation result of the sequence with the highest alignment score (Hits) was used as the final result.

## Difference analysis between different groups

The microbial composition and functional composition were analyzed using R. Results of the matrix of PCA, PCoA, and NMDS were analyzed with the vegan package. Differences between groups were obtained using Kruskal Wallis's test or Wilcoxon test, and $P$ values were corrected using the Benjamini-Hochberg method. The LEfSe method was used to analyze the differential microbes and functions.

## Fragmentation pattern analysis

According to the fastp report, the insert size of cfDNA was identified. The length of the insert size peak was used as the fragment size, and they were used as markers for classification model construction. Length of the cfDNA sequences was displayed by the density distribution curve.

## Interactions between blood and fecal microbiome

The matched samples were selected, and the corresponding abundance matrix of species was extracted. Pearson correlation coefficient and $P$ value of each species between blood and feces were calculated. Those species with $P$ values lower than 0.05 were considered important microorganisms. The interaction network was built using

Cytoscape (51), and the linear equation fitting of species in blood and fecal samples was obtained using GraphPad Prism.

## RF model for diagnosis

To select important markers for diagnosis, RF models were constructed. For the fecal microbiome, differential species and functions of LEfSe analysis were used as potential markers. For the blood microbiome, all the species and functional annotations were used as potential markers. First, the AUCRF package was used to identify the optimal combination of microbial markers. Second, the samples were divided into the training set and validation set (70:30). These markers were used to reconstruct the RF. Finally, RF models were evaluated (accuracy, sensitivity, specificity, precision, recall, and F1 values) using the caret package.

## Statistical analysis

All statistical analyses were performed using R V.3.6.0 and GraphPad Prism V.9.0.0. Differences among the three groups were obtained by the Kruskal-Wallis test, and differences between two groups were obtained by the Wilcoxon test or Student's $t$-test. $P$ value < 0.05 was considered statistically significant.

## ACKNOWLEDGMENTS

This work was supported by the Gansu Provincial Science and Technology Major Project (Grant no. 24ZDFA001), the Lanzhou Municipal Science and Technology Program (Grant nos. 2024-8-27, 2024-8-30, and 2024-4-2) and the College Students' Innovation and Entrepreneurship Program of Lanzhou University, China (Grant nos. 20240260001 and 20240260017).

P.C.: Conceptualization, Supervision, Writing–review and editing, Project administration. Z.Z.: Conceptualization, Data curation, Formal analysis, Investigation, Methodology, Resources, Software, Visualization, Writing–original draft. Y.M.: Investigation, Methodology, Resources. D.Z.: Investigation, Resources. R.J.: Investigation, Resources. Y.W.: Investigation, Resources. J.Z.: Investigation, Resources. C.M.: Resources, Validation. H.Z.: Resources. H.S.: Resources, Validation. X.J.: Resources, Validation. Y.N.: Resources, Validation. J.L.: Resources, Validation. B.Z.: Resources. L.T.: Resources. H.Z.: Resources. X.M.: Resources.

## AUTHOR AFFILIATIONS

[1]School of Pharmacy, Lanzhou University, Lanzhou, Gansu, China
[2]The Second Hospital of Lanzhou University, Lanzhou, China
[3]The First Hospital of Lanzhou University, Lanzhou, China
[4]The Third People's Hospital of Gansu Province, Lanzhou, China

## AUTHOR ORCIDs

Peng Chen http://orcid.org/0000-0003-4402-4568

## FUNDING

| Funder | Grant(s) | Author(s) |
| --- | --- | --- |
| Gansu Provincial Science and Technology Major Project | 24ZDFA001 | Peng Chen |
| The Lanzhou Municipal Science and Technology Program | 2024-8-27,2024-8-30,2024-4-2 | Peng Chen |
| The College Students' Innovation and Entrepreneurship Program of Lanzhou University, China | 20240260001,20240260017 | Peng Chen |

## DATA AVAILABILITY

All clean sequence data and sample information for this work have been deposited to the NCBI SRA database (PRJNA1086261 and PRJNA1079906).

## ETHICS APPROVAL

The study protocol was approved by the Regional Ethical Review Board of the Second Hospital of Lanzhou University, the First Hospital of Lanzhou University, and the Third People's Hospital of Gansu Province, and informed consents were obtained from each patient (21YF5FA112, 2021A-152, 2022-02-28, and 2020-07-07).

## ADDITIONAL FILES

The following material is available online.

### Supplemental Material

**Supplemental material (mSystems00276-25-s0001.docx).** Figures S1 and S2.

### Open Peer Review

**PEER REVIEW HISTORY (review-history.pdf).** An accounting of the reviewer comments and feedback.

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
