## [Reviewer comments · mSystems]

Microbiome and fragmentation pattern of blood cell-free DNA and fecal metagenome enhance colorectal cancer micro-dysbiosis and diagnosis analysis: A proof-of-concept study

Zhongkun Zhou, Yunhao Ma, Zhang DeKui, Rui Ji, Yiqing Wang, Jianfang Zhao, Chi Ma, Hongmei Zhu, Haofei Shen, Xinrong Jiang, Yuqing Niu, Juan Lu, Baizhuo Zhang, Lixue Tu, Hua Zhang, Xin Ma, and Peng Chen

Corresponding Author(s): Peng Chen, Lanzhou University

Review Timeline:

Submission Date:

February 24, 2025

Accepted:

April 2, 2025

Editor: Naseer Sangwan

Reviewer(s): Disclosure of reviewer identity is with reference to reviewer comments included in decision letter(s). The following individuals involved in review of your submission have agreed to reveal their identity: Jun Hu (Reviewer #2)

Transaction Report:

DOI: <https://doi.org/10.1128/msystems.00276-25>

Re: mSystems00276-25 (**Microbiome and fragmentation pattern of blood cell-free DNA and fecal metagenome enhance colorectal cancer micro-dysbiosis and diagnosis analysis: A proof-of-concept study**)

Dear Prof. Peng Chen:

One of the original reviewers didn't accept the request to review this revised version; therefore, I reviewed this version myself.

Your manuscript has been accepted, and I am forwarding it to the ASM production staff for publication. Your paper will first be checked to make sure all elements meet the technical requirements. ASM staff will contact you if anything needs to be revised before copyediting and production can begin. Otherwise, you will be notified when your proofs are ready to be viewed.

Sincerely,
Naseer Sangwan
Editor
mSystems

Reviewer #2 (Comments for the Author):

The author's explanation of the issues raised and appropriate modifications are acceptable